

# Mean circulation and EKE distribution in the Labrador Sea Water level of the subpolar North Atlantic.

Jürgen Fischer[1], Johannes Karstensen[1], Marilena Oltmanns[1], Sunke Schmidtko[1]

[1]GEOMAR Helmholtz Centre for Ocean Research Kiel, Düsternbrooker Weg 20, 24115 Kiel

Correspondence to: Jürgen Fischer (jfischer@geomar.de); Johannes Karstensen (jkarstensen@geomar.de);

**Abstract:** A long term mean flow field for the subpolar North Atlantic region with a horizontal resolution of approximately 25 km is created by gridding Argo-derived velocity vectors using two different topography following interpolation schemes. The 10-d float displacements in the typical drift depths of 1000 m to 1500 m represent the flow in the Labrador Sea Water density range. Both mapping algorithms separate the flow field into potential vorticity (PV) conserving, i.e. topography following contribution and a deviating part, which we define

as the eddy contribution. To verify the significance of the separation, we compare the mean flow and the eddy kinetic energy (EKE), derived from both mapping algorithms, with those obtained from multiyear mooring observations.

The PV-conserving mean flow is characterized by stable boundary currents along all major topographic features including shelf breaks and basin-interior topographic ridges such as the Reykjanes Ridge or the Rockall Plateau.

Mid-basin northward advection pathways from the northeastern Labrador Sea into the Irminger Sea and from the Mid Atlantic Ridge region into the Iceland basin are well-resolved. An eastward flow is present across the southern boundary of the subpolar gyre near 52°N, the latitude of the Charlie Gibbs Fracture Zone.

The mid-depth EKE field resembles most of the satellite-derived surface EKE field. However, noticeable differences exist along the northward advection pathways in the Irminger Sea and the Iceland basin, where the

deep EKE exceeds the surface EKE field. Further, the ratio between mean flow and the square root of the EKE, the Peclet Number, reveals distinct advection-dominated regions as well as basin interior regimes in which mixing is prevailing.

**Keywords:**

- Subpolar North Atlantic
- Intermediate depth EKE
- Subpolar mid-depth advection
- Argo data, OceanSITES data
- Ocean model metric





## 1 Introduction

The subpolar North Atlantic has been in the focus of both observational and modelling efforts in hindsight of circulation- and water mass changes as part of the climate relevant Atlantic Meridional Overturning Circulation (AMOC; reviewed e.g. by Daniault, et al., 2016). In this context the intermediate depth circulation, which also determines the spreading pathways of newly ventilated Labrador Sea Water (LSW) through the subpolar North Atlantic (SPNA), is of specific importance and has been investigated from observations and models for several

decades. A better understanding of the mechanisms that control the transport properties at mid ocean depth through the interplay of advection and diffusion is fundamental to our understanding of subpolar LSW circulation and export, and thus potentially subpolar AMOC contributions. Unlike the surface circulation, which can be analyzed for example from satellite and drifter data, the intermediate depth circulation and energetics is known to a much lesser extent. Studies that map energetics at the intermediate depth from observational data and

at gyre scales are rare but identified for example as important evaluation metrics for basic verification of ocean model simulations, including CMIPs models (Griffies et al. 2016).

In the late 1990ies, technology of profiling floats advanced such that investigations of the intermediate deep circulation could be undertaken. Two experiments were carried out in the western subpolar North Atlantic (mainly in the Labrador- and Irminger Seas) using Profiling ALACE (PALACE) floats and are of special interest

to the investigation carried out herein. The first was by Lavender et al. (2000) with a large fleet of floats drifting through the Labrador and Irminger Seas in 700m depth (the approximate depth level of upper Labrador Sea Water in the subpolar North Atlantic). A major result of the study was that the intermediate depth circulation could well be described as a cyclonic boundary current system along the topography and a series of anticyclonic recirculation cells adjacent to the Deep Western Boundary Current (DWBC). The second experiment was

dedicated to the boundary current off Labrador, and conducted in summers 1997 and 1999 with 15 PALACE floats seeded into the DWBC off Labrador to drift at 1500m, the core depth of classical LSW (Fischer and Schott, 2002). The main finding of this study and contrary to the expectations was that none of the floats was able to exit the subpolar gyre via the boundary current route. Instead, some of the floats confirmed the existence of a recirculation cell off Labrador and others indicated an eastward route following the North Atlantic Current

at its northeastern pathway. This result stimulated a series of Lagrangian experiments (Bower et al., 2009) using RAFOS drifters but also model studies (e.g., Spall and Pickart, 2003).

With the deployment of the global array of Argo profiling floats at the end of the 1990s the number and spatial homogeneity of displacement vectors at the floats parking depth of typically 1000 m or 1500 m increased

significantly. The data set is assembled in the YoMaHa'07 data base (Lebedev et al., 2007). Based on this much larger database it is of interest to revisit the earlier results. One of the immediate questions is how robust the earlier findings are, and moreover whether the present day Argo data coverage would be sufficient to proof and possibly refine the earlier results. There are two approaches to these objectives, one is to investigate temporal changes of the deep circulation on interannual time scales but with a drawback on spatial resolution. Palter at al.,

(2016) followed that approach and found a slowdown in boundary currents flow in the Labrador Sea but no significant changes in the large scale subpolar gyre circulation. Another approach, and this is taken here, is to neglect temporal variability and use all available displacement data for determining a mean flow field on a finer



spatial resolution that resembles narrow circulation elements in higher resolution compared to what have been discussed in the past.


Several attempts have been undertaken to estimate advective (long term mean) and diffusive contributions in the displacement vectors on the basis of statistical and physical constraints. While the displacements of the profiling floats may be well suited to determine the long term mean of the flow field, this is not straight forward for the eddy component of the flow field (Davis, 2005). The author suggested calculating the diffusivity from

displacement anomalies **u'** calculated from the difference of the mean flow **<U>** and the measured displacement vector $\mathbf{U_m}$. Here, we loosely follow the method proposed by Davis (1991), in which the mean flow is controlled by topography (f/H, with H is the water depth), an assumption that should hold true in the SPNA regime where weak stratification und small vertical current shear is encountered. Thus, we will estimate the advective part of the flow that is related to the concept of potential vorticity conservation (LaCasce, J.H., 2000), and the residual

flow contribution that is attributed to the diffusive part of the flow. Validation of this principle has been performed in the past (see Fischer and Schott, 2002; Fischer et al., 2004) through a comparison of deep displacements along curved topography in relation to moored (Eulerian) records.

We focus here on the SPNA north of 45°N and make use of the extended set of Eulerian (current meter

moorings) and Lagrangian (floats) observations available in the region. Over the previous two decades (regionally even longer) an impressive observing effort has been undertaken north of 45°N on the (intermediate) deep flow. Boundary currents are, thanks to their strength, the prominent circulation features in the SPNA and found all along the shelf edges in particular on the western side of the gyre. However, there are also interior circulation features of both advective- and eddy-dominated pattern, and the primary research objective of this

effort is to discriminate the mean flow **<U>** from the turbulent (eddy) component **u'** of the flow field from which the deep EKE field could be determined.

The paper is structured like follows: First, we briefly describe the methods to separate <U> and an accompanying **u'**, obtained for each displacement vector from the difference between the observed displacement

and the displacement projected to a PV contour. The fields obtained by two different gridding methods are verified for internal consistency, and in comparison to independent measurements from mooring records. Next, a gridded velocity and an Eddy Kinetic Energy (EKE) field of relatively high spatial resolution (order 25km grid size) for the SPNA is created by both gridding procedures. We discuss the fields for internal consistency based on major flow features. Furthermore, the ratio of advective flow and diffusion (Peclet number) is estimated. The

EKE field at depth is then compared with the EKE field at the surface, based on satellite data. The gridded data sets are provided for download and further use e.g. for model/data comparison, so far we are not aware of an intermediate depth EKE map.

## 2 Material and Methods

Two quality controlled Argo displacement (deep and surface) sets exist, but cover somewhat different time spans. Here, we use the Yomaha07 - Argo dataset (Lebedev et al., 2007) which contains estimates of velocities



of deep and surface currents using data of the trajectories from displacements between consecutive dives of Argo floats. This data set is updated frequently on a monthly basis.

### 2.1 Temporal and spatial distribution of the Argo float array

By March 2017 (this is the latest data considered for this analysis) the displacement data set includes data from 4284 floats stored in nine Data Assembly Centers (DAC's) worldwide and about 297,000 values of velocity. We define a velocity vector as the displacement between an Argo float descent (last surface position) and the consecutive ascent (first surfacing position) divided by the corresponding time difference. Some inhomogeneity in position and time accuracy based on the communication and positioning technology (ARGOS, Iridium) is discussed in Lebedev et al. (2007). The nominal position of the velocity vector is the mean position between the descent and ascent position pair.

The area under investigation ranges from 45°N, the latitude just south of Flemish Cap, to 65°N, which is just south of Denmark Strait (Figure 1). The westernmost longitude is 62°W, i.e. the Labrador shelf break, and to the east, the area is bounded at 7°W west of the British Isles. The evolution of the Argo data in this domain shows a rapid increase in data density in the first 5 years of the program (**Figure 2**), and from 2006 onwards the data density is adding around 2500 to 3000 displacement vectors per year; this is roughly equivalent to the number of T/S profiles gained through Argo per year. Maximum annual data increase is reached in 2011/12 with 4000 additional current vectors each year. Thereafter, the yearly data gain stabilizes at 2500 to 3000 current vectors. The regional data density ranges from approximately 10 to more than 100 per bin (2°longitude x 1° latitude; **Figure 1);** the bin size corresponds with the typical area in which data will be used for the interpolation to a certain grid point. Given the barotropic nature of the flow field in the SPNA we merged the displacement vectors from the two drift depths 1000m and 1500m depth. Considering the temperature and salinity data recorded by the floats the mean potential density field at 1500 m varies between $\sigma_\theta = 27.72$ kg m$^{-3}$ and 27.92 kg m$^{-3}$ with an average density of $\sigma_\theta = 27.77$ kg m$^{-3}$. This density is slightly lower than the commonly used lower boundary of classical LSW at $\sigma_\theta = 27.80$ kg m$^{-3}$ and thus, the resulting circulation pattern represents the core depth of the LSW.

### 2.2 Auxiliary data

To estimate contours of constant potential vorticity (PV) we used the high resolution topographic data 2-minute Gridded Global Relief Data (ETOPO2) (National Geophysical Data Center, 2006). This topography is based on a combination of depth soundings and depth estimates from multiple sources and gridded to a 2-minute special resolution. Only in one case we use the higher resolved ETOPO1version, but this did not change the results.

Furthermore, we used altimetry-based absolute dynamic topography from which the surface geostrophic flow and EKE were derived [Le Traon et al., 1998]. The altimeter products were produced by Ssalto/Duacs and distributed by Aviso with support from Cnes (http://www.aviso.altimetry.fr/duacs/).

Lastly, Eulerian time series data from moored instrumentation that recorded in the depth interval considered in the analysis here (1000-1500m) were used to locally evaluate the results of the gridded data product (see table 1 for an overview). Given the floats inherent sampling at 10 days, the moored records were smoothed accordingly.

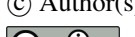



### 2.3 Separating mean flow and its fluctuation and interpolation of the results

Two interpolation methods were used to map the displacement vector data: the first is a weighted Gaussian interpolation (GI) and the second is an optimum interpolation (OI) procedure. Both methods use the same physical constraints, and both operate on an identical grid of 0.5° longitudinal range and 0.25° latitudinal range.

#### 2.3.1 Gaussian interpolation method

The strategy of the GI-method was to include two constraints in the interpolation procedure, namely a weighted distance between target (grid) point and data point, and the second is to reduce the influence of data points located in regions with very different water depth. The latter is a manifestation of our assumption that flow in the region follows PV-contours. Thus, data points across the boundary current at steep topography would only weakly be influenced from nearby but much deeper/shallower locations outside the boundary current (a topography following mapping).

The weights used have a Gaussian shape described by two parameters for each dimension: for the distance weighting we chose 40 km for the half width of the Gaussian and 80km for the cut-off – such that points outside a radius of ~80km around a selected grid point will not be used. For the other dimension (water depth difference between data location and target location as a measure of PV difference) we chose 200 m half width and 600m cut-off range. The choice of these values was guided by the dimensions of the boundary current along steep topography (e.g., the Labrador shelf break), with the width of the DWBC (Zantopp et al., 2017) between 100 km and 150 km, and a change of water depth across the DWBC from about 1000m to 3000 m. Through this procedure Boundary Currents would be conserved and not smeared out, while in the basin interior with flat bottom the weight is more toward distance – with only little influence of the depth-difference.

We analyzed the impact of different weights over a wide range of scales, but the selection applied here appears to generate the most robust result with a clear definition of the circulation elements described hereafter. Using a higher resolved grid (smaller scales) result in noisier flow field with larger overall variance, while a coarser grid (larger interpolation scales) result in a smoother field and certain details of the flow field are suppressed. The procedure could be applied to both, irregular target locations, and regular grid locations.

In a first processing step we separate the measurements into a mean flow contribution **<U>** and a fluctuating part **u**' that will be used later to determine the EKE field. Around each measurement location we selected all data within the cut-off radius and by using the selected weights (see above) we estimate a mean flow vector at the measurement location by applying the above described algorithm. Thus, we generate a velocity field that has the dimension of the original data set, and it contains only the weighted, PV-related ensemble-mean contribution (**Figure 3a**). As an illustration how well the PV-constraint works we show three floats that were deployed at roughly the same location in the northern Iceland Basin at water depth around 1800m to 2500m. This depth range is shaded all around the basin and the close correspondence of the float trajectories and the shaded area is an indication of the PV following nature of the deep flow field. By subtracting the mean component, which we assume to be inherent in each of the measured displacement vectors, from the original data a current residual (**u'**) could be calculated (**Figure 3b**).



Subsequently we applied the mapping procedure to the measured velocity field ($U_m$) to obtain a field on a regular 0.5° longitude x 0.25° latitudinal grid and the result is **<U>** on a regular grid, which is considered as one of our final data products.

The Eddy kinetic Energy (EKE) is estimated independently for each of the two interpolation methods (GI and OI). Assuming that the separation of the measured displacement vectors into **<U>** (advective) and a fluctuating

(eddy) component is successfully performed by the above methods, it allows to calculate **u'** and **v'**, the fluctuating (eddy) velocity contribution of each displacement vector (**Figure 3b**). Both eddy-components show similar overall (basin wide) statistics of Gaussian shape and equal rms-values of 4.9 cm s$^{-1}$ (**Figure 4**). The second final data product is the gridded EKE produced from the **u'** and **v'** fields derived through the first interpolation step.


**2.3.2 Optimum Interpolation method**

The second procedure uses the method of optimum interpolation (OI-method), similar to the one described in detail in Schmidtko et al. (2013). Data was only mapped if the grid points have a water depth deeper than 1200m

according to the topographic data set. All data within a radius of 110km and at locations with similar water depths – less than 1000m difference – were used in the OI. Linear gradients in latitudinal direction , logitudinal direction and water depth were fitted to the data. For the covariance matrix a diagonal value of 1.5 was used as an estimate for the signal to noise ratio (see Schmidtko et al. 2013 for details). The background field used in the optimal interpolation was taken from a least squares linear and quadratic fit of the data using depth, longitude

and latitude.

The field, resulting from the OI was used as mean flow field **<U>** which was then used to compute the residual flow (**u'** and **v'**) from each displacement vector ($U_m$) by linear four point interpolation. An individual EKE value was computed for each displacement. To exclude extreme outliers an inter quartile range filter was applied, rejecting data points 2.2 times the inter quartile range above the third quartile or that range below the first

quartile. This is similar to a 99.98% standard deviation filter in case of normal distributed data. The EKE data was then mapped in an identical procedure as the mean field.

## 3 Results

### 3.1. The intermediate depth large scale circulation from a displacement vector point of view

First we inspected the GI based interpolation of <U> on the original displacement vector positions which

represents individual mean flow realizations and added a number of selected float trajectories (**Figure 3a**). The flow realizations nicely sample the different flow regimes in the SPNA and cover the boundary currents; flow associated with topographic features, such as the Mid-Atlantic Ridge; prominent flow features, such as the deep extension of the North Atlantic Current in the 'North West Corner'. Selected areas are discussed in the following.




### 3.1.1 Boundary Currents

Individual mean flow realizations sample the boundary currents and indicate the coherence of the flow along the topography. This is also confirmed by individual float trajectories. Individual floats that were released in the northern Iceland Basin, near the northernmost part of the Reykjanes Ridge (RR), follow the deep boundary current along the topographic slope of the RR south-westward. The displacement vectors indicate swift speeds of approximately 6 to 7 cm s$^{-1}$. For the selected floats it takes about 3 month to reach the first gaps in the RR and thus to enter the Irminger Basin. However, different gaps exist and control the exchange with the Irminger Basin. After crossing the RR the floats take a northward drift on the western side of RR in the boundary current that surrounds the northern Irminger Sea and downstream merge into the deep East Greenland Current (dEGC). The selected floats stayed for almost two years in the deep boundary current inshore the 1800 m isobaths before they reached Cape Farewell, the southern tip of Greenland and which is about 2500 km downstream (comparable with a mean drift speed of about 4 cm s$^{-1}$). At about the latitude of Cape Farewell, the northward flow along the western flank of the RR is on the order of 5 cm s$^{-1}$, while the southward flow along the east Greenland shelf break regionally exceeds 10 cm s$^{-1}$. The trajectories clearly show that the PV (depth) constraint on the flow is very strong and as such our gridding procedure appropriate.

### 3.1.2 Labrador Sea

The intermediate circulation in the Labrador See shows narrow cyclonic boundary circulation where the topography is steep, i.e., along the East Greenland – and Labrador shelf breaks (**Figure 3a**), while in regions with a gentler slope (e.g. northern part of Labrador Sea) the boundary current widens considerably. From the boundary current to the interior Labrador Sea the flow reveals stable but weak recirculation cells with cyclonic rotation, and the interior of these elongated cells is almost stagnant, as is also seen in time series measurements (Fischer et al., 2010;) at location K10, where the mean 1500 m-flow is 0.8 cm s$^{-1}$ northwestward, and at K9 where the mean flow is 12.5 cm s$^{-1}$ but southeastward (**Figure 5b; Table 1**).

The nearly stagnant, weakly anticyclonic rotation is observed for the area were deep convection take place. Here the water is trapped within the closed circulation in the region of strong wintertime buoyancy loss. Both, the cyclonic recirculation cells along the Labrador shelf break, and the anticyclonic interior are thus favorable for deep convection. At 1500 m depth, the lightest water is found in the central Labrador Sea and is surrounded by extremely weak anticyclonic flow. Eventually the water in the central Labrador Sea feeds the advective path around Cape Farewell thereby exporting light and weakly stratified water into the Irminger Sea. Further south, at the exit of the Labrador Sea the flow enters a very active eddy regime in a region with very variable topography – the 'Orphan Knoll' region. Here the North West Corner of the NAC and the outflow of the Labrador Sea merge and interact.





### 3.1.3 Irminger Sea

The Irminger Sea has several characteristic flow patterns in intermediate depth (**Figure 3a**) . The most pronounced feature is the deep East Greenland Current (dEGC) that exists over the whole western part of the basin. On the opposite side the Irminger Sea is bounded by the Reykjanes Ridge that is a barrier for most of the

flow beneath 1000m depth. Further south, several gaps in the ridge allow the water from the eastern basin to spill over the ridge and a northward deep boundary current forms along the western flank of the ridge. This is one source of the dEGC. A second source of the intermediate dEGC is the mid-basin current band that is fed from the Labrador Sea and extends up to 64°N where it enters the dEGC; the cyclonic circulation that this mid-basin vein forms is sometimes called the Irminger Gyre. Within the Irminger gyre a number of long term moorings have

been maintained for more than a decade to record the thermohaline evolution of the gyre center and possibly deep convection underneath the Greenland Tip Jet (e.g. Pickart et al., 2003); the moorings are nowadays incorporated in the international OSNAP program and the OOI initiative. The mid-basin current band appears to have a number of meanders which are also visible in the 1500m geopotential derived from the Argo profile data.

### 3.1.4 Iceland Basin

The Iceland Basin, which is less well investigated, has two major topographic features that influence the circulation strongly. The western limit of the Iceland Basin is the RR and that show the already discussed boundary current. At the location of the Charlie Gibbs Fracture Zone (CGFZ), which is at about 52°N, forms dynamically the southern boundary of the basin and were the circulation at the LSW depth is eastward in connection to the North Atlantic Current (NAC) supplying water towards the eastern SPNA.

Two branches of the NAC are evident (**Figure 3a**): the majority of the floats (not shown) drift far eastward in a strongly meandering current band until arriving at the topography (still at the latitude of the CGFZ, i.e., 52°N). Thereafter the flow follows the topography northward into the Rockall Trough west of Ireland. On the western flank of the Rockall Plateau a narrow eastern boundary current forms and flows northward until it reaches the Iceland-Scotland-Ridge, where it feeds the southwestward boundary current (discussed above) that eventually

becomes a 'western' Boundary Current along the Reykjanes ridge. However, the broadest inflow comes from the mid-basin flow regime extends from the NAC northward from about 27°W, and follows the deep trench northward to 62°N. This mid-basin flow is characterized by stable advection and several large wavenumber meanders.

### 3.1.5 The North Atlantic Current regime

The southern exit of the Labrador Sea is the region where the NAC meets the DWBC (**Figure 5a**) , and while the LSW follows the topography inside a topographic feature called 'Orphan Knoll' (50°N, 46°W), the NAC is located seaward of 'Orphan Knoll' and retroflects toward east in a feature known as the 'Northwest Corner' (NWC). The latitude of the NWC is also at 52°N, and from there the NAC meanders eastward through the CGFZ. The zonal flow field and the associated southern signature of the Polar Front can be interpreted as the

southern limit of the SPNA and it forms the zonal component of the large scale cyclonic circulation. In the LSW depth range the Polar-Front separates the lighter water to the south from the denser subpolar gyre.



### 3.2 Gridded Mean Flow

By application of the GI-method, the velocity field was interpolated to a regular grid of 0.25° latitude and 0.5°
longitude (**Figure 5a**). As for the raw data maps (**Figure 3a**) the gridded data reflects all the major circulation
elements. The interpolation method keeps the deep Boundary currents as narrow- and stable jets which are
resolved by five or more grid points. Mid basin jets in the Irminger Sea and the Iceland Basin appear as
continuous but meandering pathways of the intermediate deep circulation. The correspondence of the current
field and the potential density at 1500m depth is evident. The strongest density gradients are associated with the
western boundary current elements along the eastern Reykjanes Ridge, associated with the EGC, and to a lesser
extent with the Deep Labrador Current (DLC). In combination with the deep density the major export routes for
newly ventilated LSW are also visible in the potential density pool of the central Labrador Sea draining into the
Irminger Sea. There is also a connection between the NWC and the convection area by a low density anomaly
that is not associated with the DWBC, but with the reverse circulation into the Labrador Sea.

Although, we only show the mean gridded flow field from the GI-method we do obtain the same results from the
OI method. The differences of the two estimations mainly contain small scale elements that reflect the scales of
the influence radii by either method.

### 3.3 Gridded Eddy Kinetic Energy

From the individual u' and v' fields we generated a smoothed and gridded version of the EKE (**Figure 5b**) using
the same interpolation parameters as for the mean field – i.e., both fields have the same length scales in
consideration, and the grid is identical. 'Smoothed' also means that some de-spiking and noise reduction during
the gridding operation was applied, as there were a few individual spikes along the edges of the mapping
environment, i.e. in regions where the mapping area intersects the 1500m topography and where floats might
have become bottom-stuck. These spikes could be easily detected and accounted to less than 2% of the data
contributing to an individual grid point. As a result the cleaned EKE distribution is smoother and more reliable.

We note several intense EKE hot spots in the Labrador Sea, in the 'Northwest Corner' of the North Atlantic
Current, and in the eastern SPNA located east and west of the Rockall-Plateau. While it is not surprising that the
retroflection of the NAC (i.e. the 'Northwest Corner') shows large EKE values exceeding 250 cm$^2$ s$^{-2}$, it is
surprising that the zonal basin-crossing of the NAC has relatively weak EKE at LSW levels. The second
strongest EKE is located in the northeastern Labrador Sea and is generated by instabilities and eddy shedding of
the West Greenland Current (WGC), known to occur from surface flow observations. This EKE maximum
shows relatively large values around 60 cm$^2$ s$^{-2}$ and covers a large fraction of the interior Labrador Sea. Then
there are mid-basin EKE maxima in both, the Irminger Sea and even stronger in the Iceland Basin, extending
along the whole lengths of the basins. Interestingly, there are EKE minima along both sides of the Reykjanes
ridge and zonally all across the basin just north of the NAC. Comparable weak EKE is located directly at the
topography off Labrador and off East Greenland where the DWBC is stabilized by the steep topography.

The EKE maximum in the central Labrador Sea has been linked in the past to the West Greenland Current
(WGC) (e.g. Brandt et al. 2004, Eden and Böning, 2002). For example Eden and Böning (2002) attributed the



EKE maximum to barotropic instability of the WGC with a seasonal peak at the time of maximum surface forcing (winter wind-stress maximum). It appears that there is a significant difference of the EKE intensity on both sides of the Labrador Sea; while the northward flowing WGC is subject to intense eddy formation and hence high EKE, the southward flowing Deep Labrador Current is much more stable with remarkably low EKE levels. A possible reason is the PV-conservation that stabilizes the flow when progressing southward (towards

lower f) and consequently the flow is driven toward the stabilizing topography, while for northward flow there is a tendency to move into deeper water with smaller topographic Beta. This is further supported by the weak EKE in the southward flowing East Greenland Current.

### 3.4 Advection versus Diffusion -- Peclet Number

The western subpolar basin has very different regimes regarding mean flow and EKE pattern. Even at larger

depth, there are narrow boundary currents along the topography, there are interior persistent current bands, and there are regimes of almost stagnant mean flow with intense eddy motion, but it is a priory not clear which of the processes – advection or diffusion dominates in either of the circulation regimes. This objective is investigated through the calculation of a local dimensionless number, the Peclet Number (**Pe**), which is the ratio of advection to diffusion:

$\mathbf{Pe = L_d * <U> / \kappa}$; with $\mathbf{\kappa = \alpha \sqrt{EKE} * L_d}$ ;    for EKE see **Figures 3b, 6a**;

**α** is an empiric scaling factor; here we chose **α= 0.25 ,** such that the resulting **Pe**-field varies between zero and one; $\mathbf{L_d}$ = Lagrangian length scale chosen to be related to the first baroclinic Rossby Radius (order 1 to $2*10^4$ m) ; **<U>** = is the mean current speed taken from the gridded velocity fields.

The resulting Pe-distribution ( **Figure 5c**) basically shows two regimes; one with very small **Pe** (i.e. **Pe** < 0.2),

and these are the regions where the deep eddy motion dominates the current field, such as the central basins of the SPNA with the central Labrador Sea being the largest area with low **Pe**. Similarly, the southern Irminger Sea shows low values of **Pe**, and in addition to these, the transatlantic zone south of the CGFZ is also subject to intense eddy motion.

The contrasting regimes with strong mean flow and relatively high **Pe** are the Boundary Currents along the east-

and west Greenland shelves and all along the Labrador coastline. There are also deep western and eastern Boundary currents along the Reykjanes Ridge and along the Rockall Plateau. In these areas, the advection is relatively strong compared to the eddying motion. Finally, there are mid-basin regions with stable advection and relatively weak eddy motions away from the topography and associated with the cyclonic recirculation cells. One such regime is south of Cape Farewell and it extends far into the Irminger Sea; this band connects the

'convection regime' of the central Labrador Sea with the central Irminger Sea. A similar mid-basin advection regime is found in the Iceland Basin where it connects the high **Pe**-band associated with the zonally oriented Polar Front at 52°N with the meridional current band directed along the deep Maury Channel, i.e. the central axis of the Iceland Basin. Finally, the recirculation regime off the Labrador shelf break is associated with relatively high **Pe**, as the eddy motion is relatively weak.




## 4 Verifications of the results

We verified our results in three different ways: First the results from GI and OI were compared in order to identify a superior interpolation method. Then we compared the mean flow and mean EKE fields with similar quantities derived from Eulerian time series data from moored stations ($EKE_{moor}$) in the region; and the third way
of verification was a comparison between the deep EKE and the $EKE_{surf}$ from satellite SLA data.

### 4.1 Consistency of interpolation techniques

The mean flow fields from the two gridding methods are surprisingly similar and there are no significant differences between the velocity and speed fields.. The overall speed-difference is -0.16 cm s$^{-1}$ which illustrates
that there are no systematic differences (biases) between the two speed estimates as a result of the gridding technique. The difference field are patchy in structure with patch-scales of the order of the interpolation radii. Thus, by choosing the GI method the current map (Figure 5a) is considered representative and independent of the two mapping procedures applied.

Likewise the difference in GI and OI interpolated EKE fields (Figure 6) agreed well. Most of the EKE
differences occurred in the range ± 5 cm$^2$ s$^{-2}$ with the strongest deviations around the NAC path across the SPNA – here, the GI method produces somewhat larger values. In contrast, the Northwest Corner reveals larger EKE values for the OI method. A patchy structure is observed with scales associated to the influence radii of the gridding methods (roughly 100km). The difference has an overall Gaussian distribution but with a slight bias of 1 cm$^2$ s$^{-2}$ toward larger EKE in the OI method map. The magnitude of this bias depends on de-spiking method
used in any of the two processing methods. The strongest impact is due to the removal of individual large spikes in the EKE in the GI method, which leads to a regional reduction of the corresponding EKE field. The removal of only 1% of the largest velocities results in an increased in bias to 4 cm$^2$ s$^{-2}$. No explicit de-spiking has to be used in the OI method, as it is inherent in the method itself (see Schmidtko et al., 2013), while in the GI-method we explicitly had to remove some large outliers.


### 4.2 Comparison with local Eulerian measurements

The second method for verification was a comparison between the derived mean fields (<U> and EKE) and selected locations were time series data from moored instrumentation was available (**Figure 5b; table 1**).

*4.2.1 Labrador- and Irminger Seas*

In the 'convection' area of the Labrador Sea a time series of currents is available at the K1 site since 1996 (the site is close to where the Ocean Weather Ship 'Bravo' was operated). The mooring had continuous velocity records at 1500 m depth (from 1996 to 2016) and a shorter record at 750m (from 2006 to 2016) (Figure 7). In general the mean flow at the location of K1 is very weak (order 1cm/s) with a northwestward direction into the
Labrador Sea and, given the mooring position, consistent with the anticyclonic circulation around the basin center (**Figure 5a**). Short time scales dominate the variability of the flow (**Figure 7**), and the spectra indicate


that the bulk of the energy is on intra-seasonal periods with strong decay toward longer time scales. The strongest variations occur in late spring and are associated with eddies shed by the WGC near the location of Cape Desolation (Avsic et al. 2006; Funk et al., 2009). These eddies are only weakly sheared in the LSW depth

range which is important aspect as it supports combining 1000 m and 1500 m parking depths Argo float displacements. The EKE from 180d high pass filtered time series is around 170 cm$^2$ s$^{-2}$ in both levels, 750m and 1500m (Table 1). These values are larger than what is derived from the Argo data set and we interpret this to be a result of the inherent low pass filter in the float processing. With respect to the Pe **(Figure 5c)** the area is characterized as an eddy dominated regime.

In the central Irminger Sea (CIS) a current time series is available at about 1000 m depth. As for K1, the site is characterized by a weak mean flow (around 1 cm s$^{-1}$) while the EKE (based on intra seasonal velocity fluctuations) is around 80 cm$^2$ s$^{-2}$ (Fan et al. 2013). The location of CIS is at the edge of the mid-basin velocity band connecting the Labrador Sea around the tip of Greenland, and into the Irminger Sea.

In the boundary current system a number of records could be analyzed. In general the flow is rather stable and

strong (compare also Table 1). Representative for the DWBC at 53°N (K9, Zantopp et al., 2017) the long term mean flow along the topography is 12.5 cm s$^{-1}$ and the EKE (again for periods less than 180d) is 62 cm$^2$ s$^{-2}$. Farther toward the topography the mean speed is even larger and the EKE smaller, as the DWBC appears to be more focused by the steep topography at 53°N. In any of the boundary current records a large energy contribution is on timescales less than 10 to 20 days (Fischer et al., 2015), which are not captured by the Argo

displacement vectors and different from the basin interior were the flow variability is on timescales longer than a month and thus better resolved by 10 day displacement vectors from Argo (Figure 8).

The records in the center of the DWBC at Hamilton Bank the total EKE of the moored record is larger than that from the float displacement but similar to K9 (**Table 1**). For a better comparison we calculated the EKE fraction

that Argo would represent in their 10 day displacement vectors by low-pass filtering the mooring data (10d cut-off period of the filter). Then, the EKE values coincide much better as is demonstrated by the colored mooring numbers in **Figure 5b.**

Near the offshore edge of the DWBC, at mooring K10, the flow speed is rather low, as the mooring lies in the

transition regime between the DWBC and the recirculation pathway in the upper 2000 m, while at deeper depths it is still part of the DWBC (Zantopp et al., 2017). At 1500 m the flow is mainly reverse to the DWBC direction and the EKE is rather small, but in good agreement with the EKE from Argo.

Associated with weak mean speeds (only 10% of the DWBC speed is found at locations offshore of K10) and

moderate EKE$_{moor}$ values coincide when the resulting Peclet Numbers (**Figure 5c; Table 1**) are low and indicate sufficient diffusion in the presence of weak advection. This structure is reflected in the Argo flow pattern, which shows an increasing advective contribution further toward the basin interior, and from the mean current map and the density field it is tempting to assume this route as one of the supply routes for the deep central Labrador Sea.





### 4.2.2 Subpolar Locations

Moored observations in the Iceland-Scotland-Overflow Water were available at 4 positions (Named I, S, O, W; see Kanzow and Zenk, 2014). Only three (S, O, W) moorings delivered data in the appropriate depth range for this study. While S was located in the area of low deep EKE, the fluctuations increase toward east with mooring *W* located in the EKE max along the northward flow (**Figure 5a**).

North of the I S O W array the **Iceland Array** is located at the shelf break south of Iceland, and the northern mooring direct at the topography reflects the low $EKE_{moor}$ typical for topographically guided currents, while the one further offshore is located in the northern extension of the EKE maximum of the Iceland Basin.

During the Jasin program in the late 1970s a number of moorings were deployed in the northern Rockall Trough (Gould et al., (1982)) and these moorings reflect the intermediate intensity of the deep $EKE_{moor}$ that is also present in the Argo derived values (**Table 1**).

The EKE from the moorings represent mean regional variations. In order to compare the high resolution time series with the Argo data a 10d low-pass filter is applied. There is a remaining discrepancy between EKE from Argo and from moorings with a tendency that in region with low EKE (taking now the Argo derived map as a reference) the Argo estimates are larger than the 10d-lowpass filtered mooring estimates, while in regions of high EKE the situation is reversed. We interpret this discrepancy by the inherent (nonlinear) temporal filtering in the EKE derived from Argo that tend to low-pass filter the field with an unpredictable filter characteristics (depending at which times the floats enter the corresponding interpolation radius).

### 4.3 Surface EKE versus intermediate depth EKE

In addition to the deep EKE we estimated the surface EKE ($EKE_{surf}$) field calculated from remote sensing-based ADT observations. The geostrophic surface flow from SLA contains variability over a wide range of frequencies, and some of the long term components are not generally thought to be part of the turbulent eddy field. Thus, we extracted the intra-seasonal variability by applying a high-pass filter (Hanning window) with a cut-off period at 180d. The result is a field of geostrophic fluctuations from which we calculate $EKE_{surf}$ (**Figure 8a**). This field is independently derived, and thus allows an independent comparison of the Argo-derived fields (here, the deep circulation and EKE).

The $EKE_{surf}$ also resembles major (deep) circulation elements, such that the zonal flow in the CGFZ region located underneath the zone of maximum $EKE_{surf}$ gradient at the surface. (Note in Figure 8 only currents larger than 1.5cm s$^{-1}$ are shown and thus, only vector magnitudes that would be sufficient to travel one Rossby Radius within the 10 d schedule of the floats are included). A similar surface versus deep EKE and flow pattern is seen for the northeastern flow from the Labrador Sea into the Irminger Sea. Within the Iceland Basin the deep flow is associated with the surface $EKE_{surf}$ maximum, suggesting the mid basin path is present from surface to LSW depth range. Interestingly the surface EKE shows a clear EKE minimum all along the DWBC in the western SPNA, and this is due to the slanting shape of the Boundary circulation and the slope of the western shelves. In a region with less slope, i.e. the northern Labrador Sea we observe strong EKE at all levels (surface and LSW depth range). This is the area, when the deep WGC turns away from the steep Greenlandic shelf and intense eddies are formed and shed from the DWBC (Eden and Böning 2002).





Following Ollitraut and de Verdiere (2013) we calculated the logarithmic ratio of surface EKE to the deep EKE; i.e. **ln (EKE$_{surf}$ / EKE)** , such that the ratio becomes negative when the deep EKE is larger than that at the surface. Generally, in a baroclinic ocean one would expect positive ratios, with the EKE$_{surf}$ sufficiently larger

than the EKE at depth, as is the case for the region south of the North Atlantic Drift, i.e., south of 52°N. A global much coarser map of such a ratio reveals that this is the case for almost the whole Atlantic Ocean (Ollitraut and de Verdiere; 2013). In their paper, the subpolar North Atlantic appears as broad negative area in which the deep EKE exceeds the upper layer or is of similar magnitude. The much higher resolution of the field generated herein (**Figure 8b**), allows a more detailed view, which reveals two centers of deep EKE dominance. The first is

associated with the DWBC all along the Labrador shelf break and the strongest signal around Hamilton Bank. The second center is associated with the deep action center south of Cape Farewell that shows both, stable advection and EKE at depth, while at the surface these components are rather weak. This zone extends far north into the Irminger Sea where it appears to be related to the deep EGC and its variability. This behavior indicates that for the interbasin spreading and mixing of newly formed water masses the deep EKE field contains

important information , which is not easily available elsewhere; at least not from the surface variability alone.

Besides the boundary current related anomalies there is one additional zone in which the deep EKE is close to the surface EKE, and that is along the CGFZ at the northern flank of the NAC. In this area the flow is guided by the deep topography and advection appears to be dominating the zonal flow (relatively large Pe).

**5 Summary and Conclusion**

The results of the investigation can be summarized as follows:

1) Based on nearly 17 years of quality controlled Argo displacement vectors a high resolution (~25km grid) map of mean flow in the depth layer of the LSW was constructed for the Subpolar North Atlantic. Robust circulation elements were identified consisting of boundary currents along topographic slopes,

mid-basin advective pathways, and stagnation regimes with very low mean speeds.

2) The mapping procedures were twofold: Gaussian Interpolation (GI) and Optimum Interpolation (OI), both methods were applied using potential vorticity constraints, and the resulting mean flow fields were very similar – almost identical.

3) The second product was the fluctuating (eddy -- u', v') velocity component, which was determined as

the residual after subtracting the average and potential vorticity conserving contribution from the individual measurements (displacement vectors). The u', v'-fields were used to map the mean EKE-distribution; to our knowledge for the first time.

4) The ratio of mapped mean flow to the square root of the EKE, the Peclet-Number (Pe), was estimated and showed regions that are advection dominated (boundary currents and internal LSW routes), and

regions with low PE, in which eddy diffusion prevails.

5) The mapped fields were analyzed for consistency between the OI and GI method. In addition velocity time series from moored sensors were used to estimate mean flow and EKE in an attempt to verify the mapped fields locally with independent data. While the general pattern of high and low EKE regimes are consistent, but the mooring EKE appears to be larger than EKE from Argo but the differences





535 become smaller, when the Eulerian measurements are lowpass filtered with a cut-off at the Argo sampling time scale (10d).

 6) Comparing the mid depth EKE with the independently derived surface EKE from Aviso SLA-data, we found qualitative agreement of the two fields in many regions, with the surface EKE larger than the mid depth EKE. However, other regions showed the local EKE maxima were horizontally displaced

540 between surface and the deep EKE, thus there are areas with and with larger EKE at mid depth. This seems to be a special (robust) feature of the subpolar North Atlantic.

The gridded velocity field can be used for a variety of follow up investigations, e.g. estimating water mass

545 spreading via artificial tracer release experiments or using the gridded flow field as a reference level velocity for geostrophic calculations (e.g. based on Argo derived geostrophic shear).

By focusing on the Labrador Sea, the "surprisingly rapid spreading" of LSW throughout the subpolar North Atlantic (Sy et al., 1997) is well supported by our gridded mean flow field: newly formed LSW is exported by the mid-basin advective pathway into the Irminger Sea (**Figures 3a and 5a**) and eastward through the pathway

550 that connects the western SPNA with the northern Iceland Basin through the NAC and its northern pathway. Individual floats released in the DWBC off Labrador used that path to drift within a few (3-4) years far north into the Iceland Basin.

More regional aspects were discussed in the float release experiments performed in the late 1990, i.e. before Argo started officially. On the basis of these investigations, export pathways for LSW out of the Labrador Sea

555 were discussed (e.g. Straneo et al., 2003) in which the export of LSW into the Irminger Sea, and the Boundary Current Export around Flemish Cap were identified as major export routes. While the Irminger Sea route appears strong and robust, the flow along the topography (Flemish Cap and Grand Banks) is relatively weak. Instead, the second major export route is into the eastern SPNA via the NAC route.

Traditionally the upper ocean eddy variability represented by the EKE distribution has been investigated from

560 SLA data (Brandt et al., 2004, Funk et al. 2009). Just recently (Zhang and Yan, 2018) the Labrador Sea surface EKE based on altimeter data has been investigated with regard to interannual to decadal variability in the time period 1993 to 2012. They find strong interannual variability in the EKE field near the WGC, but no trend over the observational period.

Generally, mid depth EKE maps based on observational data are rare but important for the deep ocean water

565 mass and tracer spreading. Thus, both the mean current field and the EKE at the transition between the deep water masses LSW to LNADW should be useful metrics for ocean model evaluations.

## 6 Data availability

The raw data is available through open access from: Yomaha, Aviso, Coriolis Data center. The data products

570 derived herein will be made freely available with the publication. The data set will contain gridded





(latitude/longitude grid) versions of velocities and eddy kinetic energies alongside with water depth at grid location.

**Author contribution**: Jürgen Fischer prepared the manuscript with contributions from all co-authors, all authors worked on the analysis of the data: Johannes Karstensen in general and on moored records; Marilena Oltmanns
on Argo profile data, and Sunke Schmidtko applied the OI method.

**Acknowledgements:** This project has received funding from the European Union's Horizon 2020 research and innovation program under grant agreement 63321 (AtlantOS) and grant agreement 727852 (Blue-Action). The Argo data were collected and made freely available by the international Argo project and the national programs that contribute to it (http://doi.org/10.17882/42182). We further acknowledge the Yomaha'07 group for
generating the Argo displacement data set. This paper contains products from data supplied by Natural Environment Research Council, and from data gathered by the RACE program of the German Ministry BMBF. OOI data were obtained from the NSF Ocean Observatories Initiative Data Portal, http://ooinet.oceanobservatories.org. Mooring metadata is available via BODC and the OceanSITES network (www.oceansites.org).

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




**Figures**

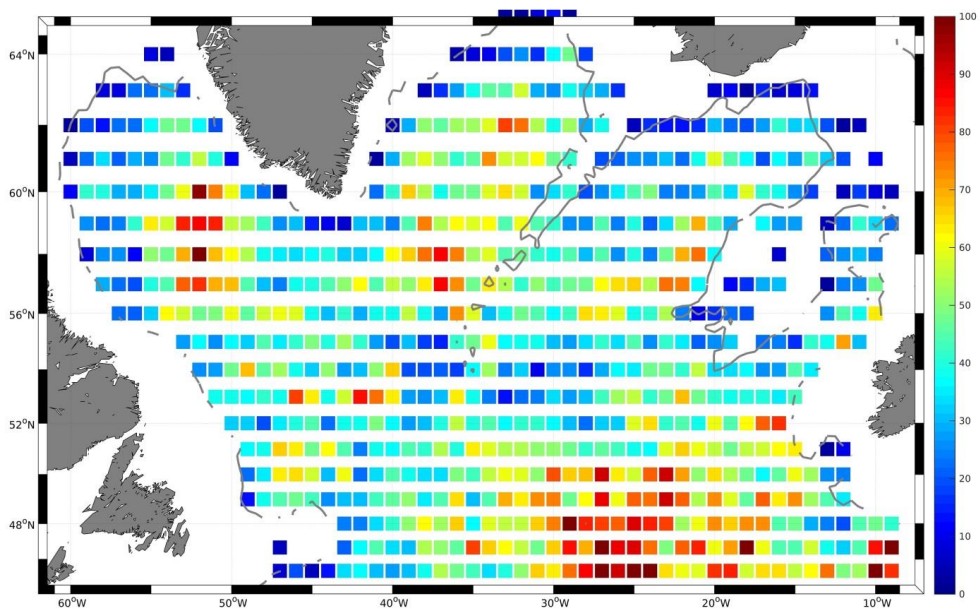

**Figure 1: Data density in 1°*1° fields; the number of 10d- displacement vectors per cell.**

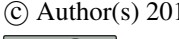



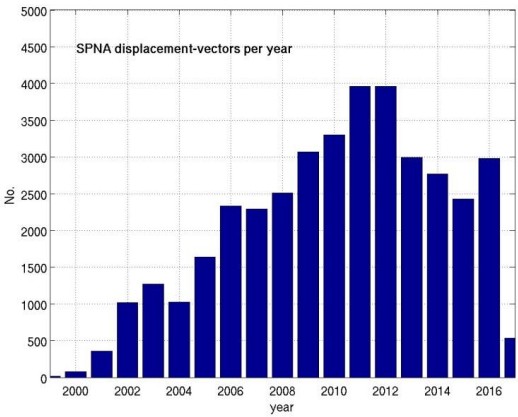


**Figure 2: Temporal evolution of the Argo data density in the subpolar domain independent of the parking depth (1000 m and 1500 m), and from year 2000 to March 2017.**





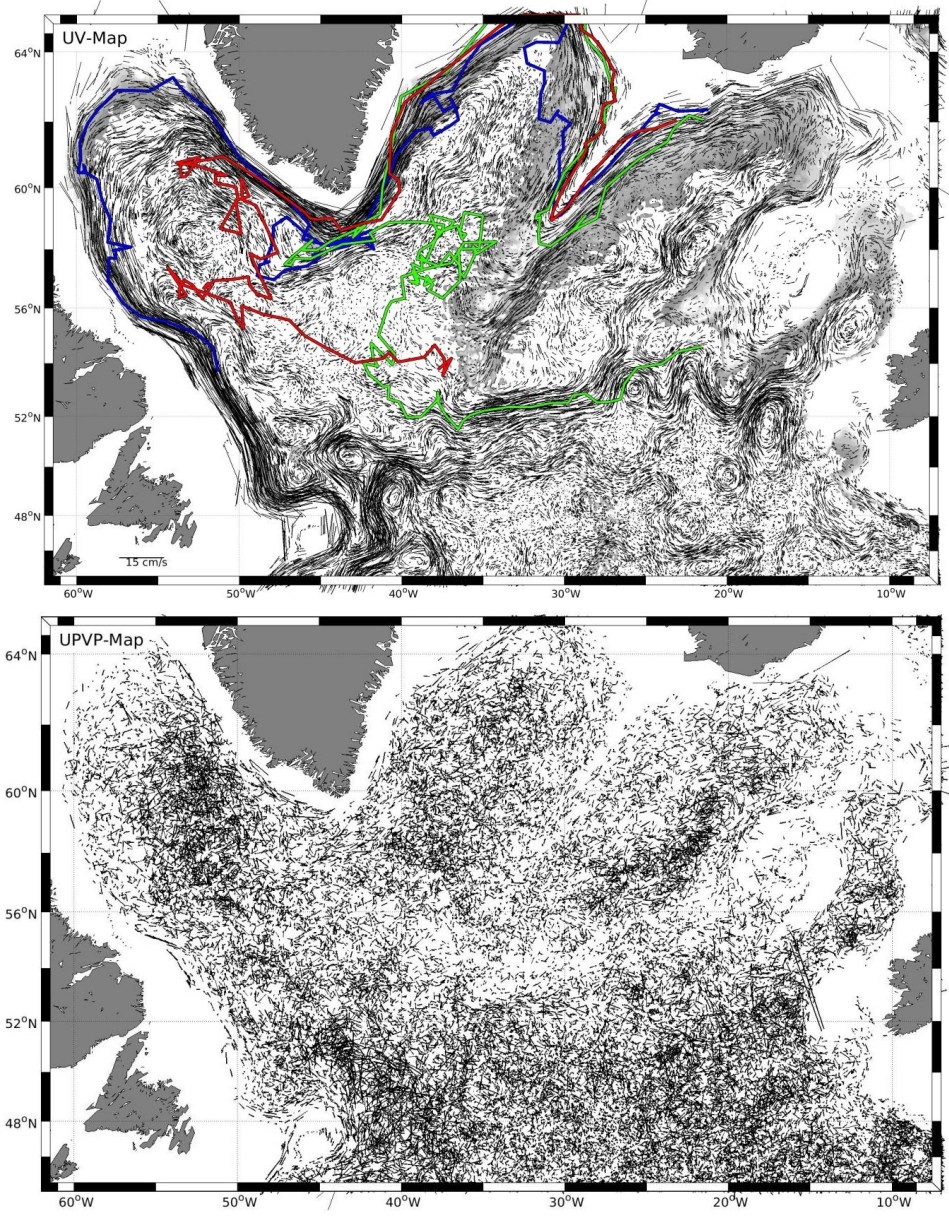

**Figure 3: Mid-depth circulation (a) in the western subpolar NA from ~38500 Argo deep drifts (1000m or 1500m parking depth) derived from the Yomaha'07 data. This is an attempt to present the advective**

**contribution of the flow field at each measurement location; i.e. for each measured 10-day drift vector (for details of the processing see text). Colored lines for selected float trajectories (deployed in the Iceland basin); grey shaded area for topographic depth range (1500m to 2500m). The residual (b) is thought to be u',v', the eddy contribution to the flow field – note, the scaling vector in (b) is a factor 2 larger than in 3a.**







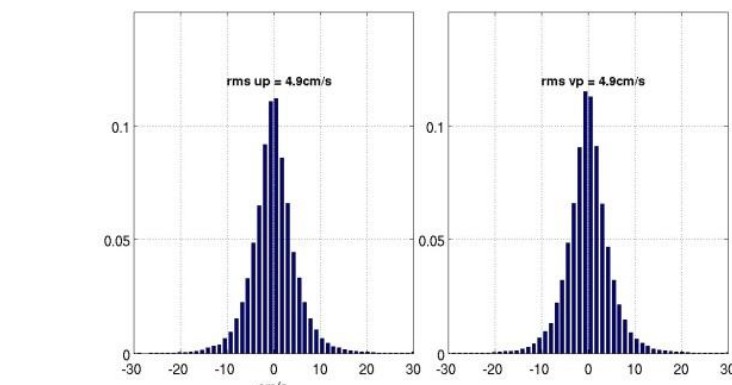

**Figure 4: Normalized distribution of the Mid-depth eddy velocity components; left is u' (east-west**
**component) and right is v' (north-south component) – Gaussian distribution with equal rms of 4.9 cm s⁻¹**





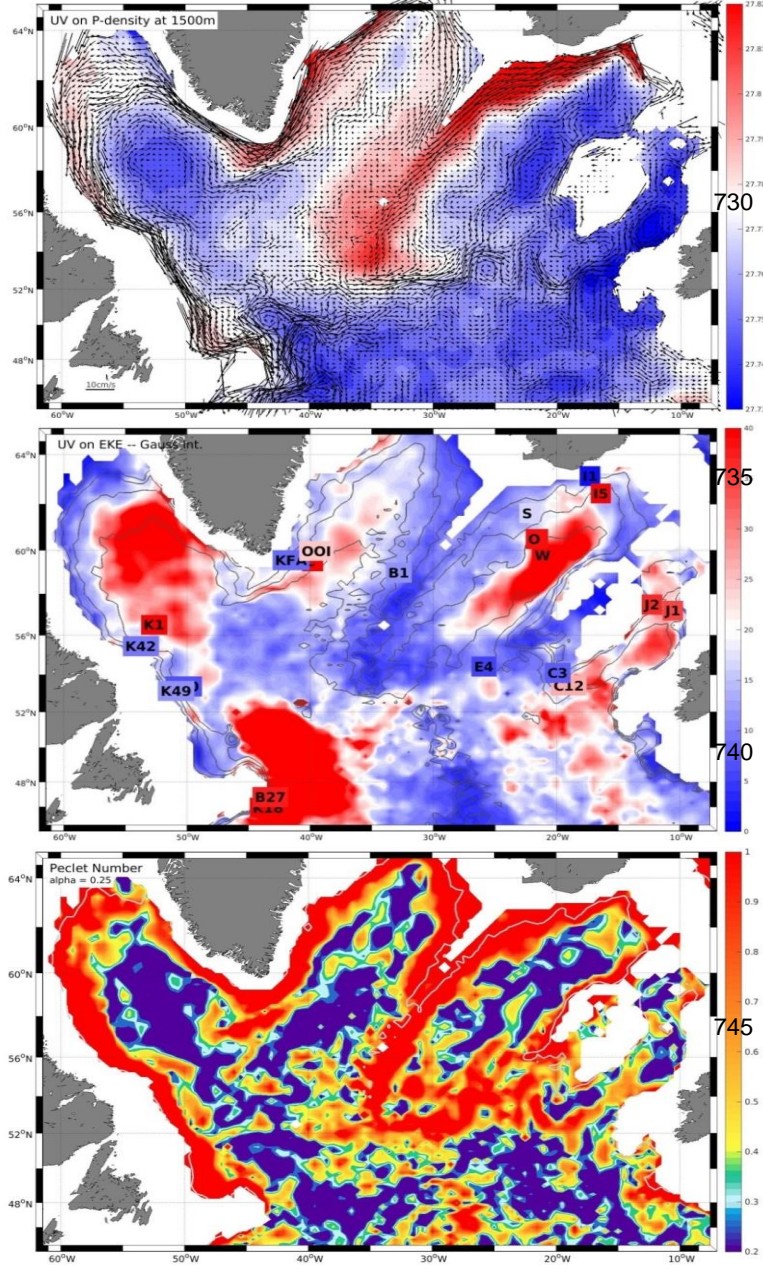

**Figure 5 a) Gridded velocity field from the GI method overlaid on potential density distribution from 1500 m depth. b) Gridded EKE (in cm² s⁻²) map from the GI method with selected EKE values from moored observations (numbers in boxes). Mooring fluctuations are lowpass filtered at 10d cut-off for better comparability of mooring time series with 10 day displacement velocity from Argo data. Mooring location markers are colored with respect to the EKE from the moored record. c) The ratio of mean speed to the square root of the EKE scaled by a factor alpha=0.25, i.e. a measure of the Peclet number Pe.**



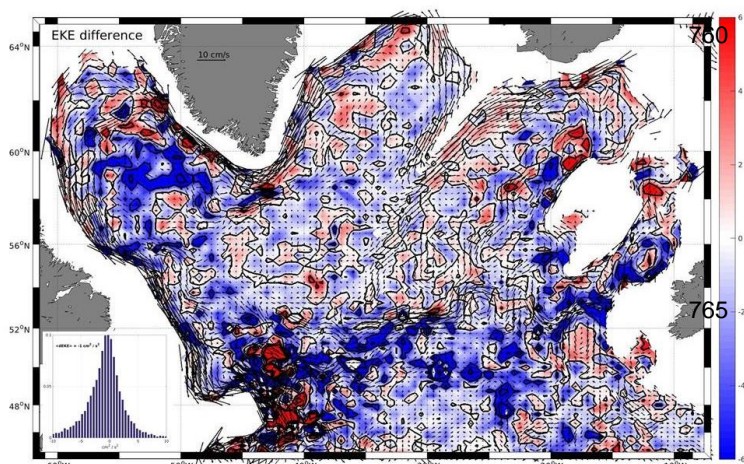

**Figure 6: EKE-Map Difference between Optimum Interpolation and Gaussian Interpolation method. Inlet**
**at lower left: Histogram of Difference reveals Gaussian shape and a weak bias of <dEKE> = -1 cm$^2$ s$^{-2}$**






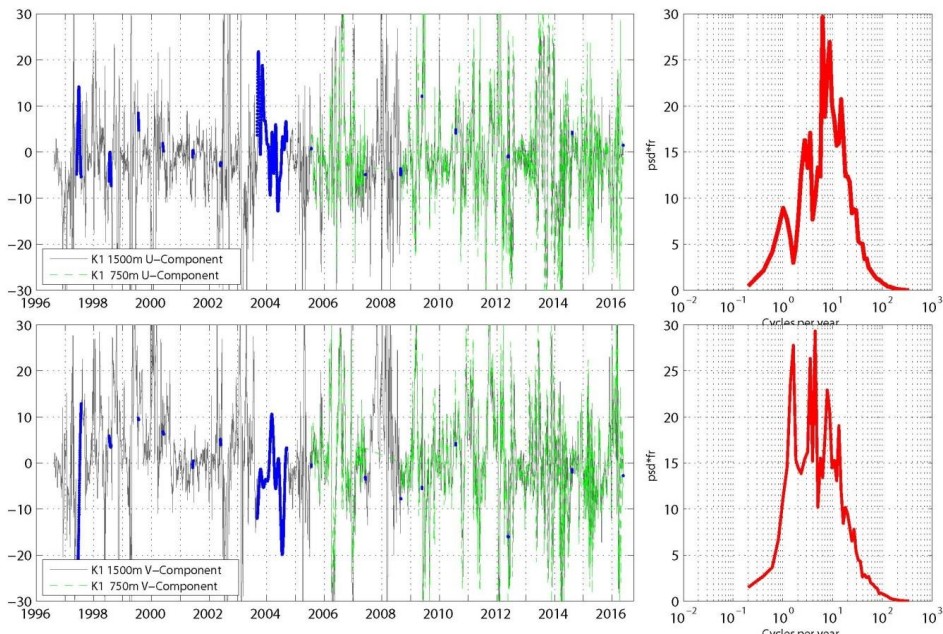


**Figure 7: Example of a current time series from the central Labrador Sea at mooring K1. Two depth levels were occupied regularly (750m since 2006 (green curve); 1500 m since 1996). Gaps (blue lines) are filled by EOF based interpolation (Zantopp et al., 2017). High frequency spectra from 1500 m records (right).**






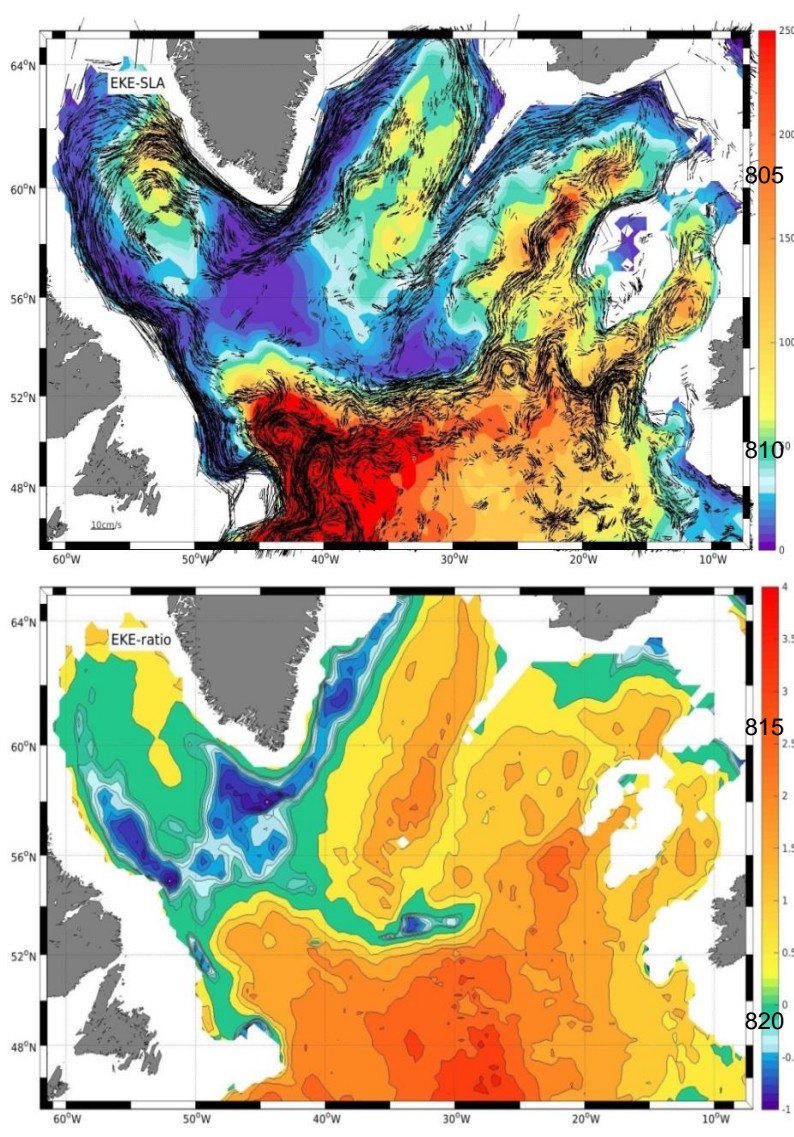

**Figure 8: a) Surface EKE derived from the AVISO geostrophic surface flow which is high pass filtered at**
**180d cut-off (in cm² s⁻²) as an estimate of the geostrophic turbulence. Overlaid is the Argo derived mean**
825 **(PV-related) flow at 1000-1500m depth, with flow speeds below 1.5 cm s⁻¹ omitted; this better reveals the**
**major advective pathways. b) The logarithmic ratio of surface EKE to deep EKE; green and blue colors**
**show areas in which the deep EKE dominates.**



830 **Tables**

**Table 1: Eulerian EKE - statistics in the Subpolar North Atlantic. BODC: British Oceanographic Data Centre (www.bodc.ac.uk)**

| Mooring Nominal instrument depth | latitude | longitude | $\langle Spd \rangle$ cm s$^{-1}$ | EKE$_{Full}$ cm$^2$ s$^{-2}$ | EKE$_{10dlp}$ cm$^2$ s$^{-2}$ | EKE$_{GI}$ cm$^2$ s$^{-2}$ | Pe | Comment |
|---|---|---|---|---|---|---|---|---|
| **Moorings in the North West Atlantic** | | | | | | | | |
| K42$_{1500}$ | 55° 27.5'N | 53° 43.8'W | 16.1 | 29 | 6 | 18 | 6.6 | AR7W mooring |
| K49$_{1500}$ | 53° 08.5'N | 50° 52.1'W | 12.5 | 39 | 12 | 20 | 3.6 | Records from the |
| K10$_{1500}$ | 53° 22.8'N | 50° 15.6'W | 0.2 | 13 | 7 | 13 | 0.1 | 53°N observatory |
| | | | | | | | | |
| K1$_{1500}$ | 56° 31.5'N | 52° 39.0'W | 1.6 | 165 | 72 | 34 | 0.2 | Mid Basin moorings |
| CIS$_{1000}$ | 59° 42.7'N | 39° 36.2'W | 1.6 | 37 | 21 | 20 | 0.3 | |
| | | | | | | | | |
| OOI | 59° 58.5'N | 39° 28.9'W | 1.9 | 63 | 25 | 20 | 0.4 | OOI Irminger Sea (access |
| | | | | | | | | |
| K18$_{1500}$ | 46° 27.1'N | 43° 25.1'W | 4.3 | 78 | 50 | 52 | 0.6 | Flemish Cap |
| B227$_{1100}$ | 47° 06.2'N | 43° 13.6'W | 27.3 | 60 | 38 | 50 | 4.3 | Moorings |
| B1$_{1534}$ | 59° 48.5'N | 32° 48.5'W | 2.1 | 18 | 12 | 13 | 0.6 | Reykjanes Ridge (access via BODC) |
| KFA | 59° 35.0'N | 41° 33.0'W | | 18 | 9 | 12 | | Cape Farewell, NOCS (access via BODC) |
| | | | | | | | | |
| **Moorings in the North East Atlantic** | | | | | | | | |
| I3$_{1135}$ | 62° 43.1'N | 16° 49.2'W | 6.1 | 110 | 26 | 16 | 1.2 | Iceland Array |
| I5$_{1403}$ | 62° 26.4'N | 16° 28.3'W | 3.1 | 90 | 54 | 21 | 0.4 | (access via BODC) |
| S$_{1245}$ | 61° 04.1'N | 22° 11.5'W | 4.5 | 47 | 17 | 16 | 1.1 | ISOW transport |
| O$_{1480}$ | 60° 30.5'N | 21° 36.1'W | 4.0 | 68 | 39 | 24 | 0.6 | Array |
| W$_{1520}$ | 59° 46.8'N | 20° 56.6'W | 1.0 | 133 | 90 | 48 | 0.1 | |
| | | | | | | | | |
| J1 | 57° 12.9'N | 10° 34.0'W | 3.2 | 41 | 35 | 23 | 0.5 | Jasin moorings |
| J2 | 57° 30.1'N | 12° 16.0'W | 3.3 | 44 | 37 | 22 | 0.5 | (access via BODC) |
| C3$_{1290}$ | 54° 05.2'N | 19° 55.0'W | 5.6 | 15 | 8 | 17 | 2.0 | Conslex moorings |
| C12$_{1260}$ | 53° 25.2'N | 19° 18.0'W | 2.8 | 35 | 27 | 23 | 0.5 | (access via BODC) |
| E4 | 54° 24.8'N | 25° 54.1'W | 3.1 | 7 | 6 | 8 | 1.3 | WOCE mooring (access via BODC) |