# Peer review of "Mean circulation and EKE distribution in the Labrador Sea Water level of the subpolar North Atlantic."

_Ocean Science, 2018_

## Referee Comment (RC1) · Anonymous Referee #1 · 12 Jul 2018

This is a review of "Mean circulation and EKE distribution in the Labrador Sea Water level of the subpolar North Atlantic" by Fischer et al. on Ocean Science Discussions.

This manuscript is primarily a technical demonstration of fine resolution gridding of Argo trajectory data at mid-depth. It uses the YoMaHa07 trajectory dataset (based on differencing surfacing positions, divided by the time between surfacings) to derive an estimate of the 10-day mean velocity at the Argo float parking depth (typically 1000 m or 1500 m). The authors use two gridding procedures which preferentially grid data along f/H contours which, in the high latitudes of the Labrador Sea/subpolar North Atlantic means gridding across a longer distance along isobaths than across isobaths.

[Figure]

They use all available Argo float trajectories over the Argo period to date to create an average circulation in the subpolar gyre, and then look at the residuals to this average circulation to create a gridded estimate of mid-depth EKE. The scientific analysis is limited to estimate a Peclet number (ratio of advection to horizontal diffusion, i.e., ratio of the mean velocity map to the EKE map) and note areas where advection dominates or diffusive processes are expected to be large. The main new findings of this paper are the remarkable maps of the high resolution currents including the expected strong boundary currents against topography, but also the notional pathways of the North Atlantic Current crossing the Atlantic from west to east, and northwards in the eastern subpolar gyre. It also shows some remarkable eddy activity which would be difficult to observe in a statistically robust fashion any other way. While the final calculation of the Peclet number is somewhat simplistic, I believe the paper is worthy of publication and will be of interest to a great many oceanographers concerned with the Atlantic in this region of climate sensitivity and climate forcing.

The methods and approach are valid and justified, though the authors could provide more detail on the underlying dataset (YoMaHa) particularly with regards to any biases that may occur during the profiling time of the floats. While the time spent profiling is much less than that spent at parking depth, in much of the world's oceans, this can have a marked influence on the overall trajectory determined as velocities tend to be surface intensified. This may be less of an issue in the subpolar North Atlantic where velocities are more barotropic. I think this would be straightforward to show using the mooring data available to the authors (which was also used to validate their mapped product).

The presentation quality is of high standard, and well written, save for a few comments on the text noted below by line number.

The paper has relatively few references for a region that is so well studied, both observationally and numerically. Many of the boundary currents seen in the mean maps are well-established. I would recommend referring the reader to a few papers on these,
perhaps when going through the description of the currents in section 3.1. There are several summary papers that could be used to catch all in a small handful of references.

In summary, this is a lovely technical demonstration of Schmidtko's gridding procedures applied to an underutilized dataset (the YoMaHa trajectories). It is timely, given the recent new observational efforts in the subpolar gyre (OSNAP), and the publication of the dataset and derived velocity/eke fields may be useful to researchers trying to understand both the horizontal circulation in the subpolar gyre and it's role in the AMOC. The result is a remarkable fine resolution picture of the mean circulation in the subpolar gyre (particularly Fig 8) as well as the identification of a few deep regions of EKE which are likely important to the spreading of the properties in the DWBC more broadly in the subpolar gyre. I recommend this paper for publication in OS after minor textual revision.

L57 1990ies -> 1990s

L77 proof -> prove

L100 structured like -> structured as

L121 Yomaha07 -> YoMaHa07 (and later in the paper)

L151 define potential vorticity. f/H?

L183 "with only little influence of the depth-difference." awkward. I suggest "with little influence of the underlying bathymetry" or "with less influence due to the water depth difference"

L186 result in noisier -> results in noisier

L187 result in a smoother -> results in a smoother

L188 I could not parse what was meant by "could be applied to both, irregular target locations, and regular grid locations". I suspect it is a problem of punctuation and that perhaps what was meant is "could be applied to both irregular target locations and

regular grid locations."

L196-197. There is reference to shading in the figure, but at least in my printed copy this is difficult to see. Perhaps use a pale blue shading so that the eye is not tricked into seeing clusters of thin black lines as part of the grey shading.

L198 current residual -> residual velocity (suggested)

L203 Eddy kinetic Energy -> Eddy Kinetic Energy or better "eddy kinetic energy". There are quite a few words that I don't believe should be punctuated throughout, e.g., "Boundary" in "Boundary current"

L216 remove space "latitudinal direction , longitudinal" before comma

L242 Suggest replacing "control" to "direct". I typically associate the word "control" with something dynamical, and the exchange may be influenced by topography but I don't believe it has been shown to be controlled by topography.

L258-259. The table is useful, but it would be helpful to have the moorings called out (e.g., K10) with their lat/lon in the text so that the reader does not need to use the text, then the table then the figure to see the area that is being referred to.

L260 rotation -> circulation, were -> where

L251 remove comma after Both

L252 remove comma after shelf break

L264 Quantify how weak "extremely weak" is? What flow speeds are observed here?

L267 Provide lat/lon for Orphan Knoll

L282 Write out acronyms on first usage, e.g. OSNAP and OOI. Suggest also including a reference for at least OSNAP (perhaps Lozier et al., 2017)

L288 were -> where

L297 Can you quantify a wavenumber/wavelength even if by eye? Looks to me to be about 7 degrees or roughly 700 km

L339 remove comma after both

L341 Comparable -> Comparably

L351 Define beta, and don't capitalise

L354 larger -> greater

L355 depth -> depths

L355 and 356, remove second and third instance of "there are"

L356 a priory -> a priori

L359 Specify what you are calling diffusion here. I expect that it is EKE as a sort of horizontal diffusion, but this can be specified (and referenced)

L361 empiric -> empirical

L369 don't capitalise "boundary currents", also L371

L409 were -> where

L420 which is important -> which is an important

L435 were -> where

L567 Provide web links for the three sources: YoMaHa, Aviso and Coriolis DAC.

Fig 3. I don't know what the labels within the figures mean "UV-map" seems straightforward, but perhaps Velocity or mean velocity map. But for 3b "UPVP"? This is perhaps u'v'?

Fig 8. These figures are striking and beautiful. It is a shame that the vectors in panel A cannot be more clearly seen over the chosen colormap. Can you un-saturate the blue

end of the color scale so that the boundary currents in the Labrador Sea and around Greenland are clearly visible?

---

## Referee Comment (RC2) · Anonymous Referee #2 · 18 Jul 2018

A long term mean flow field for the subpolar North Atlantic region with a horizontal resolution of approximately 25 km is created by gridding Argo-derived velocity vectors using two different topography-following interpolation schemes. The deviation from the topography-following component is interpreted as the eddy contribution. The results from these procedures compares favorably against EKE calculated from a number of long-term moorings in the region. With the circulation field thus verified, the authors then interpret different dynamical regions by comparing the mean advection against the eddying contribution, via computation of a Peclet number. The result of this comparison is reasonable: advection is more dominant in boundary currents, eddies relatively more dominant in the sluggish interior of the basins. Finally, the manuscript compares the

deep (nominally 1000 m EKE) against surface EKE from satellite altimetry, and show that there are several regions where deep eddies appear to be more energetic than surface eddies.

The manuscript is well written, and many of the figures are beautiful. There is broad interest in the circulation of this region, and the statistics of the deep EKE advances the field. The interpretation of the key results is well supported, except in a couple of places of concern that I outline below. I believe the paper is worthy of publication following attention to a small number of major points, and some additional minor details.

Major:

I was suprised that the authors chose to merge the 1000 m and 1500 m float displacements (Line 143) without doing any thermal wind adjustment. A back-of-the-envelope calculation suggests that the assumption that flow is barotropic enough to justify the lack of attention to this step is questionable. For instance, the manuscript states that the density varies by 0.2 kg m-3 at 1500 m. If this density change occurs over a distance of 100 km or less (not unreasonable according to Figure 5a), the velocity will change by more than 10 cm s-1 over the 500 m separating the 1000 m and 1500 m displacements. So, by averaging without accounting for this difference, you might bias your estimate of the mean flow and the eddies.

Second, I am a little bit hesitant about the interpretation of the Peclet number. Specifically, it is not clear how Ld chosen. This choice will strongly effect the result. If a spatially-varying Ld is chosen (as might be appropriate, given varying latitude, stratification, and ocean depth), this should be described. If a constant is used, this should be clear and also justified. Related to this point, I do not see clear evidence that eddies must dominate below Pe < 0.2. The stated justification for alpha = 0.25 is not physical, and there is uncertainty in Ld, so it seems that this Pe has quite a bit of uncertainty. Overall, I think the calculations should be better explained. I also think that this discussion should be more about relative strength of eddies versus advection, rather than
which dominates in an absolute sense. The low Pe regions are certainly locations of weak mean flow, where the eddying is a more important part of the momentum budget than the high Pe regions.

Finally, it is not clear which altimetry product was used. If it is the gridded product, it should be noted that it has much lower EKE than that produced from an along-track product (Zhang and Yan 2018). Thus, the comparison of the surface and deep EKE might have fewer regions with deep EKE > surface EKE. The overall interpretation of Figure 8 need not change, but I believe a note regarding what size eddies the altimeter product resolves is in order.

Minor

Line 45 - Not sure what is meant by "hindsight"

Line 77 - "proof" -> "prove"

Line 190 - I don't understand this sentence, which seems critical to the method. Some equations would help.

Line 194 - I don't see how the three float trajectories plotted on top of the mean flow field is an indication of how well the PV-constraint works

Line 215 "All data within a radius of 110 km and at locations with similar water depths – less than 1000m difference – were used in the OI." How was 1000 m chosen? Were there any sensitivity tests that informed this decision?

Is Figure 3 made using the gridding procedure with a penalty in the cross-isobath direction? It would be helpful to point toward the exact subsection in the text where plotting procedure is described in the Figure caption. If this procedure is used, then it seems circular to argue that the coherence of the velocities along isobaths is evidence that this procedure is appropriate (e.g. Line 249).

Line 290 - It is not clear what is "not shown."

Line 314ff - Make a reference to Figure 5a to guide the reader to the appropriate figure.

Line 356 - A priory -> a priori

Line 395ff - The manuscript goes through the exercise of comparing the two mapping methods (GI and OI). In this comparison, it comes to light that de-spiking to remove the top 1% of the largest velocities as part of the GI method, increases the bias relative to the OI method by 400%. This seems like a cautionary tale for users of each method, but the authors stop short of explaining how to avoid biases from anomalous velocities. It would be useful for the authors to give a more extended judgement on the promises and pitfalls of each method.

Line 410-412 - quotes are unnecessary around 'convection' and 'Bravo'

Line 438 - no verb

Line 540 - "thus there are areas with and with larger EKE at mid depth." Seems to be a word missing.

Line 555 - This study does not look at flow at the Grand Banks, and southward flow around Flemish Cap appears strong in Figure 3 and 5a, with relatively high Pe in Figure 5c. Therefore, I do not understand the evidence provided in support of this conclusion: "While the Irminger Sea route appears strong and robust, the flow along the topography (Flemish Cap and Grand Banks) is relatively weak." I suggest removing this sentence or including some references to support it.

Figure 5 - I assumed the color of the boxes used the same colorscale as the background contour map, but this should be spelled out in the caption. Colorbar and axis labels all too small

---

## Author Comment (AC1) · 13 Aug 2018

Autor Response 1: We first would like to thank the reviewer (1) for the detailed review and the very encouraging remarks to this manuscript. We appreciate the recommendations in form and content; which we generally accepted – this has led to a considerable improvement of the paper. Although; the reviewer in his final statement requests only minor textual changes, we feel some of the comments are too important not to be discussed in more detail. Regarding the comments pertaining to the figures, we followed the recommendation to brighten the colormap in Figure 8, such that the contrast of the vector field is enhanced and the circulation is better visible. We further changed the

internal labels in Figure 3 to mean velocity and to eddy velocity and now use a blue color shading for the topographic slope along the steep areas of the bathymetry.

The reviewer asked for some substantial alterations which we will discuss in the following:

RC1 a) The reviewer identified the derivation of the Peclet Number as the only scientific contribution of the paper.

AR1 a) In fact, we feel that the estimation of a mid-depth EKE field is the major finding and has important scientific consequences for the distribution of mid depth water masses in circulation system with sufficiently different regimes of advection and diffusion at mid depths. The Pe, as it is estimated here, should be seen as a relative (qualitative) quantity that allows to detect regional differences in the advection/diffusion distribution at mid-depth (also recommended by reviewer 2).

RC1 b) The reviewer requested more details about the underlying dataset (YoMaHa) particularly with regard to biases that may occur during the profiling time of the floats.

AR1 b) Sources of uncertainties are in fact manifold, and we elaborated on that subject in the paper and extended that discussion. We also included the underlying technical report (Lebedev et al., 2007) in which some of the error sources are also discussed.

RC1 c) The reviewer requested to directly put the geographical locations inside the text, even when listed in the Table.

AR1 c) We agree that there should be better guidance to some of the locations, but we feel just giving lat/lon data are less illustrative than for example a more descriptive note: (mooring CIS located in the center of the Irminger Sea). However, this is for mooring locations that are part of the Table and included are geographical locations. For other locations like OWS Bravo and 'Orphan Knoll' we included lat/lon in the text as recommended.

RC1 d) The reviewer requested some additional references. He argued that such a

well documented region requires some additional references of both observational and numerical studies.

AR1 d) Thus, we added several additional citations.

---

## Author Comment (AC2) · 15 Aug 2018

We first would like to thank the reviewer (2) for the detailed review and the very constructive remarks to this manuscript. We appreciate the recommendations in form and content, which we generally accepted – this will lead to a considerable improvement of the paper.

In addition to the textual remarks, the reviewer raised several substantial remarks, which we will thoroughly address.

RC2:The reviewer was surprised that the authors chose to merge the 1000 m and 1500

m float displacements (Line 143) without doing any thermal wind adjustment, and the concern would be a bias in both the mean and the eddy field if such an adjustment is not done.

AR2: Indeed we had a discussion about a thermal wind adjustment earlier in the writing phase of the manuscript, and we re-discussed this concern with respect to the remarks of reviewer 2. We came to the following conclusions: for various reasons, discussed below, we will not use any geostrophic adjustment, but we agree that more details are needed to convince the reader that joining the data from the two depth levels is appropriate. Thermal wind shear adjustment of individual data (displacement vectors) would need synoptic measurements of T/S profiles in the vicinity of the data point. Argo trajectory data of individual floats are synoptic, but would only support the cross – component, and with the concept of f/H following flow this would correspond to cross-bathymetric flow; near the topography this is generally small compared to boundary current speeds. In order to estimate the perpendicular velocity component one would need simultaneous T/S profiles from near-by floats. Within time scales of days to month this is rarely the case. The second possible version is to map the T/S field and calculate the mean shear on a similar grid. Based on the high resolution MIMOC climatology the effect is small in the western SPNA, and it is only the southeastern SPNA where the shear is stronger due to the presence of the more baroclinic NAC. Thermal wind adjustment based on this mean T/S-field will mix time and space variations in an arbitrary manner and can only be applied to the mean velocity field. In consequence it would not be possible anymore to separate the raw data inot u' and  components.

However, there is other information that may be used as justification for combining the data from both levels. First, we inspected current shears from moored records from both, the boundary current regime (Fischer et al., 2010, for mean velocity profiles in the DWBC in the Labrador Sea; Fischer et al., 2015 for the DWBC at various place around the SPNA) and interior flow (Figure 7 of this manuscript). From the Argo data we performed the gridding procedures for both depths independently, but with lower

resolution and larger interpolation radii. The fields from the 1000m floats resemble that of the final product and there are no large scale biases (basin wide means). This is different for the 1500m floats where the data coverage is smaller and there are several large gaps. In general, the flow field along the topography will be different because the shallower level between 1000 and 1500m depth is only represented by the 1000m field. There were some areas in which the data density appeared large enough in both fields; e.g. southern Labrador Sea and Irminger Sea (see attached figure 1):

For the Irminger Sea, an area with boundary currents, interior advection, and sluggish circulation, we find rather similar circulation patterns in both levels. Beside the two levels we include the velocity difference of the 1000m level minus the 1500m velocity (right subplot of figure 1) with the largest differences in areas of low data density, e.g. along the DWBC off East-Greenland. We obtained the following statistical values (see table) with the individual means being much larger than the mean difference. The standard deviation of individual components are also larger than the standard deviations of the difference field. This supports the conclusion that a statistical significant difference is not detectable and that we may combine the two fields into a single layer without any adjustment.

Field Mean (cm/s) Standard Dev (cm/s)

U1000 -0.90 2.45

V1000 -0.55 3.17

U1500 -0.78 2.86

V1500 -0.64 3.24

U10-U15 -0.12 1.60

V10-V15 0.09 1.77

RC2: The reviewer has some concern about the interpretation of the Peclet number as

a quantitative measure of advection vs. diffusion.

AR2: We totally agree on this point and would also see the Peclet number as it is defined here in a rather simplistic way as a qualitative rather than a quantitative measure. Thus we interpret the regional differences in the Pe distribution as regional variations of the relative importance of advection and diffusion. This is now made clearer in the manuscript.

RC2: Finally, it is not clear which altimetry product was used. If it is the gridded product, it should be noted that it has much lower EKE than that produced from an along-track product (Zhang and Yan 2018). Thus, the comparison of the surface and deep EKE might have fewer regions with deep EKE > surface EKE. The overall interpretation of Figure 8 need not change, but I believe a note regarding what size eddies the altimeter product resolves is in order.

AC2: We used the gridded altimetric product which has a horizontal resolution of $0.25°$ – comparable to the resolution of the grid used herein. However, as the typical eddy size in the subpolar area is of similar size (10 to 50 km), the EKE estimates are certainly biased low compared to estimates of EKE from the along-track records. Thus, the ratio of the top to deep EKE should not be seen as an absolute measure, but for identifying regional differences between deep and surface EKE fields. This is now better explained in the text.

RC2: Line 194 - I don't see how the three float trajectories plotted on top of the mean flow field is an indication of how well the PV-constraint works

AC2: The trajectories show that individual floats stay for a long time (up to years) in a narrow bathymetric depth range which is indicated (figure 3a) through the colored topographic range. While the Coriolis force (f) does not change much along the trajectories this behavior is an indication that the floats indeed follow the bathymetry and therefore f/H as a measure of the large scale potential vorticity.

RC2: All data within a radius of 110 km and at locations with similar water depths – less than 1000m difference – were used in the OI." How was 1000 m chosen? Were there any sensitivity tests that informed this decision?

AC2: The rational behind the choice of the cut of radii was twofold: first, across the boundary currents the bathymetric slope is of the order of 2000m and we would not like to smear out the boundary currents by more than half of its width. This was similar to the parameters chosen for the Gaussian interpolation (GI) method. For the GI we performed some variational tests with the interpolation scales including the cut-off radii. The chosen parameters were a compromise between resolution and smoothness of the resulting fields.

RC2: Is Figure 3 made using the gridding procedure with a penalty in the cross-isobath direction? It would be helpful to point toward the exact subsection in the text where plotting procedure is described in the Figure caption. If this procedure is used, then it seems circular to argue that the coherence of the velocities along isobaths.

AC2: This is an important point – we used the cross isobaths penalty for generating Figure 3a, and based solely on this field it is in fact circular to conclude on along isobaths coherence. However, this coherence still exists when the cross isobaths penalty is removed; although the boundary currents are becoming wider through spreading towards the basin interior. This is now stated in the text.

RC2: Line 395ff - The manuscript goes through the exercise of comparing the two mapping methods (GI and OI). In this comparison, it comes to light that de-spiking to remove the top 1% of the largest velocities as part of the GI method, increases the bias relative to the OI method by 400%. This seems like a cautionary tale for users of each method, but the authors stop short of explaining how to avoid biases from anomalous velocities. It would be useful for the authors to give a more extended judgement on the promises and pitfalls of each method.

AC2: In fact any of the mapping procedures requires some despiking of the velocities

(especially the eddy component has some spikes close to the bathymetry, see Figure 3b). These are treated differently in the respective method. While the OI method uses least square techniques in which individual spikes have little influence, the GI method uses the data more directly, and for grid points close to the spike position these get a strong weight, such that the EKE is biased low. The magnitude of this bias depends on details of the editing method. When a simple sort and remove procedure is used then the bias increased from 1 cm2 s-2 to 4 cm2 s-2. Adding additional statistic constraints (e.g. removal only values exceeding 2-standard deviations within the area used for the interpolation — order 100km range) would reduce the bias to the range of 1-2 cm2 s-2. However, we agree that a quantitative interpretation of the EKE and local variations of it should be made cautionary. This extended description is now incorporated in the text.

RC2: - This study does not look at flow at the Grand Banks, and southward flow around Flemish Cap appears strong in Figure 3 and 5a, with relatively high Pe in Figure 5c. Therefore, I do not understand the evidence provided in support of this conclusion: "While the Irminger Sea route appears strong and robust, the flow along the topography (Flemish Cap and Grand Banks) is relatively weak." I suggest removing this sentence or including some references to support it.

AC2: This has been rewritten with a not so rigid statement regarding the Flemish Cap pathway, and we added the reference (Schott et al., 2004) that has a focus on the circulation along the Grand Banks.

Fischer, J., Karstensen, J., Zantopp, R. J., Visbeck, M., Biastoch, A., Behrens, E., Böning, C. W., Quadfasel, D., Jochumsen, K., Valdimarsson, H., Jónsson, S., Bacon, S., Holliday, N. P., Dye, S., Rhein, M. und Mertens, C. (2015) Intra-seasonal variability of the DWBC in the western subpolar North Atlantic. Progress in Oceanography, 132 . pp. 233-249. DOI 10.1016/j.pocean.2014.04.002.

Schott, F., Stramma, L., Zantopp, R. J., Dengler, M., Fischer, J. und Wibaux, M. (2004) Circulation and deep water export at the western exit of the subpolar North
Atlantic. Journal of Physical Oceanography, 34 . pp. 817-843. DOI 10.1175/1520-0485(2004)034<0817:CADEAT>2.0.CO;2

[Figure]

[Figure]

**Fig. 1.** Irminger Sea circulation in depth layers